# Fast and Reliable Evaluation of Adversarial Robustness with Minimum-Margin Attack

## Abstract

The *AutoAttack* (AA) has been the most reliable method to evaluate adversarial robustness when *considerable* computational resources are available. However, the high computational cost (e.g., *100 times* more than that of the project gradient descent attack) makes AA *infeasible* for practitioners with limited computational resources, and also hinders applications of AA in the *adversarial training* (AT). In this paper, we propose a novel method, *minimum-margin* (MM) attack, to fast and reliably evaluate adversarial robustness. Compared with AA, our method achieves comparable performance but *only costs 3%* of the computational time in extensive experiments. The reliability of our method lies in that we evaluate the quality of adversarial examples using the margin between two targets that can precisely identify the most adversarial example. The computational efficiency of our method lies in an effective *Sequential TArget Ranking Selection* (STARS) method, ensuring that the cost of the MM attack is independent of the number of classes. The MM attack opens a new way for evaluating adversarial robustness and contributes a feasible and reliable method to generate high-quality adversarial examples in AT.

## 1 Introduction

The *deep neural network* (DNN) has attracted a large number of researchers from different disciplines such as computer science (Goodfellow et al., 2016; Castelvecchi, 2016; Vaswani et al., 2017), physics (DeVries et al., 2018; Huang et al., 2019; Levine et al., 2019), biology (Maxmen, 2018a;b; Webb, 2018) and medicine (Hao et al., 2015; Esteva et al., 2017). The success of DNN mainly lies in its ability to learn useful high-level features from abundant data (Deng & Yu, 2014; LeCun et al., 2015). These learned features have been successfully used to address many difficult tasks. For example, DNNs can recognize images with high accuracy comparable to human beings (LeCun et al., 1998; Krizhevsky et al., 2012). In addition, DNNs are also widely used for speech recognition (Hinton et al., 2012), natural language processing (Andor et al., 2016), and playing games (Mnih et al., 2013; Silver et al., 2016).

As the impacts of DNN increase fast, its reliability has been a key to deploy it in real-world applications (Huang et al., 2011; Kurakin et al., 2017). Recently, a growing body of research shows that DNNs are vulnerable to adversarial examples, i.e., test inputs that are modified slightly yet strategically to cause misclassification (Szegedy et al., 2014; Nguyen et al., 2015; Kurakin et al., 2017; Carlini & Wagner, 2017a; Finlayson et al., 2019; Wang et al., 2019; Zhang et al., 2020b; Gao et al., 2021). The existence of such adversarial examples undoubtedly lowers the reliability of DNNs. Meanwhile, researchers have also been considering finding a reliable way to evaluate adversarial robustness of a DNN before deploying it in the real world.

The high-level idea of evaluating adversarial robustness of a DNN is quite straightforward, i.e., generating adversarial examples and calculating the accuracy of the DNN on these examples (this kind of accuracy is also known as *adversarial robust accuracy*). Szegedy et al. (2014) first pointed out the existence of adversarial examples and used a less powerful box-constrained L-BFGS method to generate them. Based on the studies in (Szegedy et al., 2014), Goodfellow et al. (2015) put forward the *fast gradient sign method* (FGSM). One common loss function they used is *cross-entropy* (CE) loss, and to maximize the loss function, FGSM uses its gradient to determine in which direction the pixel's intensity should be increased or decreased. Madry et al. (2018) introduced a simple refinement

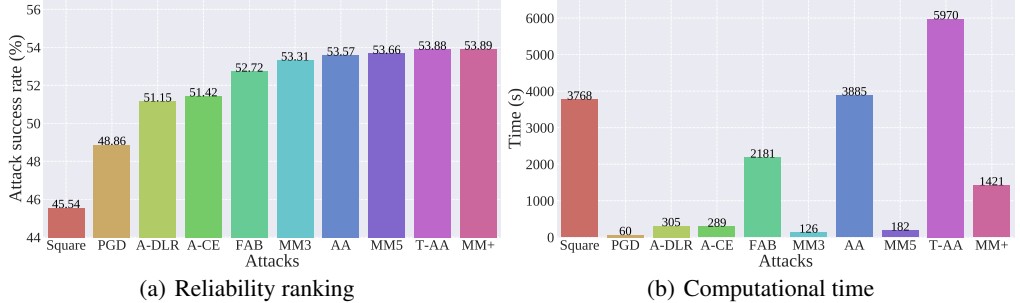

(a) Reliability ranking          (b) Computational time

Figure 1: Comparison of reliability and computational cost among baselines and different versions of MM attack. MM3, MM5 and MM+ are three versions of our MM attack. The subfigure (a) shows the (sorted) attack success rates of different attacks. The higher the success rate, the stronger the attack. The subfigure (b) shows the computational cost of these attacks. The less time, the better the attack. AA is an ensemble of APGD-CE (A-CE for short), APGD-DLR (A-DLR for short), FAB and Square. T-AA considers each target for APGD-DLR and FAB in AA, and is thus more time-consuming. Compared with AA (or T-AA), our MM3 achieves comparable performance but only costs 3% (or 2%) of the computational time. The model structure we used is ResNet-18, which follows the adversarial training of (Madry et al., 2018) on CIFAR-10.

of the FGSM: *projected gradient descent attack* (PGD), where instead of taking a single step of size $\epsilon$ in the direction of the gradient sign, multiple smaller steps are taken.

**Existing Evaluation Methods.** PGD was an effective method to evaluate adversarial robustness of a *standard-trained DNN* (Madry et al., 2018) since adversarial robust accuracy of a standard-trained DNN is always very low after using PGD. Nevertheless, the existence of adversarial examples has already inspired research on training a *robust DNN* to defend against them, which means that a standard-trained DNN is not the only DNN we might meet and we need to evaluate adversarial robustness of a *robust DNN* as well. Unfortunately, PGD fails to reliably evaluate adversarial robustness of a robust DNN (Carlini & Wagner, 2017b; Croce & Hein, 2020).

Carlini & Wagner (2017b) observed the phenomenon of gradient vanishing in the widely used CE loss for the potential failure of L-BFGS, FGSM and PGD, and replaced the CE loss with many possible choices. Croce & Hein (2020) claimed that the fixed step size and the single attack used are the causes of poor evaluations, and they put forward an ensemble of diverse attacks (consisting of APGD-CE, APGD-DLR, FAB and Square) called *AutoAttack* (AA) to test adversarial robustness. Until now, AA has been the most reliable method to evaluate adversarial robustness (Croce & Hein, 2020).

However, though AA performs well in reliability, it needs a large amount of computational time. As shown in Figure 1(a), for attacking a ResNet-18 model on CIFAR-10 (following the adversarial training in (Madry et al., 2018)), the computational cost of AA (or T-AA) is 65 times (or 100 times) more than PGD used in (Madry et al., 2018), where T-AA is more time-consuming since it considers each target for APGD-DLR and FAB in AA. Worse still, in the worst case as analyzed in Appendix A, the computational cost of AA (or T-AA) is even 109 times (or 440 times) more than PGD.

**A Dilemma Between Reliability and Computational Efficiency.** The high computational cost makes AA infeasible when considerable computational resources are unavailable. Unfortunately, such scenarios are common in the real world, e.g., for practitioners who need real-time evaluation at each epoch of the training process of a robust model, such high computational cost is unacceptable. Similarly, since a large number of adversarial examples need to be generated at each epoch during *adversarial training* (AT), such high computational cost hinders applications of AA in AT. In consideration of the high reliability but low computational efficiency of AA, and the high computational efficiency but low reliability of PGD, we seem to encounter a *dilemma*: we *have to* consider giving up one factor (reliability or computational efficiency) when evaluating the adversarial robustness.

**Our Reliable and Fast Solution.** In this paper, we are dedicated to achieving reliability and computational efficiency simultaneously. For reliability, we evaluate the quality of adversarial examples using the margin between two targets for precisely identifying the most adversarial example. For computational efficiency, we put forward an effective *Sequential TArget Ranking Selection* (STARS) method to ensure that the cost of the MM attack is independent of the number of classes.

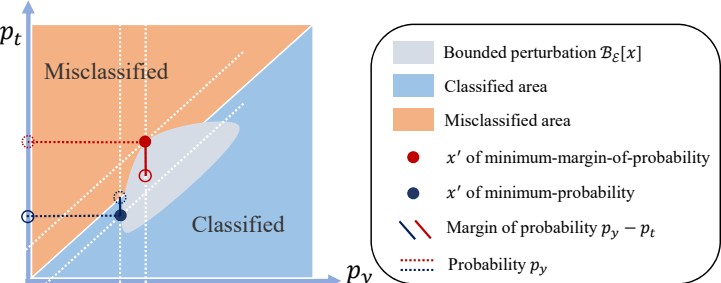

Figure 2: Minimum margin of probability. $p$ denotes the predicted probability, $p_y$ and $p_t$ are the predicted probability on the true label $y$ and a targeted false label $t$. The gray shape is the image of the adversarial variants $x'$ within the bounded perturbation ball $\mathcal{B}_\epsilon[x]$ under the mapping of the network onto $(p_y, p_t)$; the orange area $(p_t > p_y)$ indicates the region where the adversarial variants are misclassified, or to say a successful attack, while the blue area $(p_t < p_y)$ indicates the region where the adversarial variants do not attack successfully.

**Reliability.** To achieve reliability, we investigate the reasons behind the failure of PGD. We identify that CE loss, which is based on the probability of the true label $p_y$, is not an appropriate measure to the quality of adversarial examples. In Figure 2, we provide a simple demonstration to this issue, in which we consider one targeted false label $t$. As we can see, it is much more reasonable to measure the quality of adversarial examples in terms of the *margin of probability* $p_y - p_t$. The most adversarial example in Figure 2 then corresponds to the one with the *minimum margin of probability* instead of the *minimum probability* $p_y$. Detailed study of the rationality of minimum-magrin is provided in Section 3.1. Since the search space $\mathcal{B}_\epsilon$ of high dimensional images is large (grey area), previous studies use gradient descent methods to generate the adversarial example that maximizes the loss function (Goodfellow et al., 2015; Carlini & Wagner, 2017b; Madry et al., 2018).

Though it looks promising to generate adversarial examples via minimizing the margin of probability, we find that there are still two issues: (a) The probability $p$ is a kind of rescaling method to the logits $z$. Croce & Hein (2020) heuristically rescaled the logits $z$ using their proposed DLR (defined at Eq.(6)). We investigate the performance of different rescaling methods in Section 3.1. We numerically find that the method using natural logits $z_y - z_t$ (the meaning of margin) with no rescaling performs the best; (b) For the problem of multi-class and untargeted attacks, the *margin* is then $z_y - \max_{i \neq y} z_i$. However, $-(z_y - \max_{i \neq y} z_i)$ is not an appropriate loss function, because the $\max$ function only considers one target at the current step while those unconsidered targets may lead to more adversarial examples. Hence, the reliable method is to minimize $z_y - z_t$ for each $t \neq y$ and take the most adversarial one, which is a widely used solution (Croce & Hein, 2020).

**Computational Efficiency.** Although running the attack for each false target is reliable, the computational cost depends on the number of classes. For datasets with a large number of classes, e.g., CIFAR-100 (100 classes) and Imagenet (1,000 classes), the computational cost will increase accordingly. To achieve computational efficiency, we propose a *Sequential TArget Ranking Selection* (STARS) method in Section 3.2 to make the computational time independent of the number of classes. According to the ranking of predicted probability, STARS method only selects a few highest targets and runs a sequential attack. Experiments show that, benefited from STARS method, MM attack can save 76.36% of the computational time on CIFAR-10, 98.51% on CIFAR-100 and 77.78% on SVHN.

By taking all the above factors into consideration, we propose a novel method, the *minimum-margin* (MM) attack. Its detailed realization is provided in Section 3. We present extensive experimental results in Section 4, which verify that our MM attack can fast and reliably evaluate adversarial robustness. In particular, MM attack achieves comparable performance but *only costs 3%* of the computational time compared with the current benchmark AA. Hence, our proposed MM attack provides a new direction of evaluating adversarial robustness and contributes a feasible and reliable method to generate high-quality adversarial examples in AT.

## 2 PRELIMINARY

**Neural Networks.** A neural network is a function $f_\theta : \mathbb{R}^n \to [0, 1]^K$, where $\theta$ is the parameters contained in $f_\theta$ and $K$ is normally the number of classes. The output of the network is computed

using the softmax function, which ensures that the output is a valid probability vector. Namely, given an input $x \in \mathbb{R}^n$, $f_\theta(x) = [p_1, \ldots, p_K] = p$ where $\sum_{i=1}^{K} p_i = 1$ and $p_i$ is the probability that input $x$ has on class $i$. Before the softmax function, the output of the network is called logits $z$, i.e., $p = \text{softmax}(z)$. The classifier assigns the label $y = \arg\max_i f(x)_i$.

**Projected Gradient Descent Attack (PGD).** Madry et al. (2018) proposed the *projected gradient descent* (PGD) attack to generate adversarial examples to mislead a well-trained classifier $f_\theta$. Specifically, they start with setting $x^0 = x$, and then in each iteration:

$$x'_{(t+1)} = \Pi_{\mathcal{B}_\epsilon[x_{(0)}]}(x'_{(t)} + \alpha \, \text{sign}(\nabla_{x'_{(t)}} \ell(f_\theta(x'_{(t)}), y)), t = 0, 1, 2, \ldots, \tag{1}$$

where

$$\mathcal{B}_\epsilon[x] = \{x' \mid d_\infty(x, x') \le \epsilon\}, \tag{2}$$

is the closed ball of radius $\epsilon > 0$ centered at $x$; the $x^{(0)}$ refers to the starting point which corresponds to the natural example (or the natural example perturbed by a small Gaussian or uniformly random noise); $x^{(t)}$ is the adversarial example at step $t$; $\Pi_{\mathcal{B}_\epsilon[x^{(0)}]}(\cdot)$ is the projection function that projects the adversarial variant back to the $\epsilon$-ball centered at $x^{(0)}$ if necessary; the $\ell_\infty$ distance metric is $d_\infty(x, x') = \|x - x'\|_\infty$; and $\ell$ is *cross entropy* (CE) loss:

$$\text{CE}(f, x, y) = -\log(p_y) = -z_y + \log\left(\sum_{j=1}^{K} e^{z_j}\right), \tag{3}$$

where $p_i = e^{z_i} / \sum_{j=1}^{K} e^{z_j}, i = 1, \ldots, K$, and $z$ is the logits of the model outputs.

**Carlini and Wagner attack (CW).** Carlini & Wagner (2017b) observed the phenomenon of gradient vanishing in the widely used CE loss for potential failure. The gradient w.r.t $x$ in Eq. (3) is given by

$$\nabla_x \text{CE}(f, x, y) = (-1 + p_y)\nabla_x z_y + \sum_{i \neq j} p_i \nabla_x z_i. \tag{4}$$

If $p_y \approx 1$ and consequently $p_i \approx 0$ for $i \neq y$, then $\nabla_x \text{CE}(x, y) \approx 0$ in Eq. (4). This gradient vanishing issue can lead to the failure of attacks. Motivated by this phenomenon, Carlini & Wagner (2017b) replaced the CE loss with several possible choices. Among these choices, the widely used one for the untargeted attack is

$$\text{CW}(f, x, y) = -z_y(x') + \max_{i \neq y} z_i(x'). \tag{5}$$

**AutoAttack (AA).** Croce & Hein (2020) claimed that the fixed step size and the lack of diversity in attack methods are the main reasons for the limitations of previous studies. Motivated by the *line search* method (Grippo et al., 1986), they put forward *adaptive projected gradient descent attack* (APGD). They showed that using adaptive step size significantly improves the adversarial robustness compared with using fixed step size. For the loss function at Eq. (5), they claim that scale invariance w.r.t. $z$ is necessary, and they proposed an alternative loss function:

$$\text{DLR}(f, x, y) = -\frac{z_y(x') - \max_{i \neq y} z_i(x')}{z_{\pi_1}(x') - z_{\pi_3}(x')}, \tag{6}$$

where $\pi$ is the permutation of the components of $z$ in decreasing order. For the targeted attack, they propose another alternative loss function:

$$\text{Targeted-DLR}(f, x, y, t) = -\frac{z_y(x') - z_t(x')}{z_{\pi_1}(x') - \frac{1}{2} \cdot (z_{\pi_3}(x') + z_{\pi_4}(x'))}. \tag{7}$$

For the lack of diversity, they claimed that diverse attacks are beneficial for reliability, and then they put forward an ensemble of various parameter-free attacks called *AutoAttack* (AA) to test adversarial robustness, where AA contains APGD-CE, APGD-DLR, FAB and Square. *Targeted AutoAttack* (T-AA) replaces APGD-DLR with the targeted APGD-DLR, and replaces FAB with targeted FAB.

Table 1: Attack success rate (%) of different loss functions.

| Attack | Loss function | CIFAR-10 | Diff. | CIFAR-100 | Diff. | SVHN | Diff. | Tiny-Imagenet | Diff. |
|--------|---------------|----------|-------|-----------|-------|------|-------|---------------|-------|
| PGD | $-\log(p_y)$ | 48.27 | -3.09 | 73.60 | -2.92 | 41.89 | -5.67 | 78.78 | -4.19 |
| CW | $-z_y(x') + \max_{i \neq y} z_i(x')$ | 49.13 | -2.23 | 74.55 | -1.97 | 45.08 | -2.48 | 81.19 | -1.78 |
| MM | $-z_y + z_t$ | 51.36 | 0.00 | 76.52 | 0.00 | 47.56 | 0.00 | 82.97 | 0.00 |

## 3 THE REALIZATION OF MINIMUM-MARGIN ATTACK

In this section, we discuss the realization of our minimum-margin (MM) Attack. In Section 3.1, we verify the rationality of using minimum-margin as the loss function and discuss the influence of different logits rescaling methods on robustness. In Section 3.2, we propose an effective Sequential TArget Ranking Selection (STARS) method to improve computational efficiency. In Section 3.3, we provide the detailed descriptions of MM attack.

### 3.1 THE RATIONALITY OF MINIMUM-MARGIN

To understand the rationality of our minimum-margin, we first look into the situation where no adversarial attack can succeed. Then we show that the formulation of minimum-margin is naturally derived from such a situation. We say that a classifier $f$ is *completely robust* if $\forall x' \in \mathcal{B}_\epsilon[x]$, $\arg\max_i f(x')_i = \arg\max_i f(x)_i$. The following condition is necessary and sufficient to the complete robustness:

**Condition 1.** *Given a natural example $x$ with its true label $y$, the $K$-class classifier $f$ satisfies*

$$\forall x' \in \mathcal{B}_\epsilon[x], z_y(x') - \max_{i \neq y} z_i(x') \geq 0, \tag{8}$$

*where $\mathcal{B}_\epsilon[x] = \{x' \mid d_\infty(x, x') \leq \epsilon\}$; $z_y(x') = f(x')_y$; $z_i(x') = f(x')_i$.*

According to this condition, to reliably evaluate the complete robustness, the adversarial attacks should find the adversarial examples with minimum $z_y(x') - \max_{i \neq y} z_i(x')$, i.e., the most non-robust data point. It indicates that we should replace the CE loss in Eq. (3) with $-(z_y(x') - \max_{i \neq y} z_i(x'))$ as the loss function in adversarial attacks. However, as mentioned before, $-(z_y - \max_{i \neq y} z_i)$ in Eq. (5) is not an appropriate loss function since it only focuses on the current step. The reliable method is to minimize $z_y - z_t$ for each target $t \neq y$ and take the most adversarial one. To verify the rationality of minimum-margin, we conduct experiments on different datasets with $-(z_y - z_t)$ being the loss function (MM). In Table 1, MM performs a more reliable evaluation than PGD and CW.

**The comparison with Targeted-DLR.** Targeted-DLR (Croce & Hein, 2020) heuristically rescales the logits $z$ in Eq. (7). For the logit of true target $z_y$ and a false target $z_t$, the relative magnitude of $z_y$ and $z_t$ is constant under different rescaling methods. However, though the rescaling methods preserve the sign of $z_y - z_t$, they will lead to different adversarial variants. Clearly, there is no perfect rescaling method that can perform the best in any situation but inappropriate rescaling methods will hinder the searching for the most adversarial example.

We conduct experiments to investigate the difference of using different rescaling methods. The experimental setting follows (Madry et al., 2018), and we replace the CE loss with seven different logits rescaling methods. In Table 2, the successful set denotes the number of examples that can be attacked successfully in the test set of CIFAR-10 (10,000 examples). As shown in Table 2, inappropriate logits rescaling methods (e.g., Targeted-DLR in AA) reduce the reliability, and the method using natural logits (no rescaling) performs the best among the seven methods. The results motivate us to use it as the loss function. Note we do not deny that there could be a better rescaling method, but still, a reasonably good result can be obtained by no rescaling (natural logits in Table 2).

We also investigate the difference among different successful sets. The non-empty difference sets $A \cup B_i - A$ and $A \cup B_i - B_i$ in Table 2 suggest that diverse logits rescalings can be considered when considerable computational resources are available, we analyze it in Appendix C.

Table 2: The successful set of different rescaling methods.

| ID | Rescaling method | Formulation | Set size | Ranking | diff. | $A \cup B_i - A$ | $A \cup B_i - B_i$ |
|----|------------------|-------------|----------|---------|-------|------------------|--------------------|
| A | Natural logits | $-(z_y - z_t)$ | **5219** | 1 | 0 | / | / |
| $B_1$ | Softmax | $-\frac{e^{z_y} - e^{z_t}}{\sum_{i=0}^{K} e^{z_i}}$ | 5172 | 2 | -47 | 4 | 51 |
| $B_2$ | Max | $-\frac{z_y - z_t}{z_y}$ | 5165 | =3 | -54 | 5 | 59 |
| $B_3$ | Sum | $-\frac{z_y - z_t}{z_y + z_t}$ | 5165 | =3 | -54 | 5 | 59 |
| $B_4$ | Min-Max | $-\frac{z_y - z_t}{z_{\pi_1} - z_{\pi_{10}}}$ | 5121 | 5 | -98 | 3 | 101 |
| $B_5$ | DLR | $-\frac{z_y - z_t}{z_{\pi_1} - \frac{1}{2} \cdot (z_{\pi_3} + z_{\pi_4})}$ | 5078 | 6 | -141 | 2 | 143 |
| $B_6$ | Sigmoid | $-(\frac{e^{z_y}}{1 + e^{z_y}} - \frac{e^{z_t}}{1 + e^{z_t}})$ | 4820 | 7 | -399 | 1 | 400 |

---

**Algorithm 1** MM Attack

1: **Input:** natural data $x$, true label $y$, set of false labels $C$, model $f$, loss function $\ell_{MM}$, maximum number of PGD steps $N$, perturbation bound $\epsilon$, initial step size $\alpha$, the number of classes $K$, targets selection number $K_s$, checkpoints set $W$;
2: **Output:** adversarial data $x'$;
3: **while** $K_s > 0$ **do**
4:     $x'_0 \leftarrow x$; $x'_{max} \leftarrow x$; $f_{max} \leftarrow f(x'_0)$; $c = \arg\max_{i \in C} f(x)_i$;
5:     **for** $k = 0$ **to** $N - 1$ **do**
6:         $x'_{k+1} \leftarrow \Pi_{\mathcal{B}_\epsilon[x]}(x'_k + \alpha sign(\nabla_{x'_k} \ell_{MM}(f(x'_k), y, c)))$;
7:         **if** $f(x'_{k+1}) > f_{max}$ **then** $x'_{max} \leftarrow x'_{k+1}$; $f_{max} \leftarrow f(x'_{k+1})$;
8:         **if** $k \in W$ **and** (Condition 2 **or** Condition 3) **then** $\alpha \leftarrow \alpha/2$; $x'_{k+1} \leftarrow x'_{max}$;
9:     **end for**
10:    $C \leftarrow C \setminus \{c\}$;
11:    **if** $\arg\max_{i \in C} f(x')_i \neq y$ **then** $K_s \leftarrow 0$;
12:    $K_s \leftarrow K_s - 1$;
13: **end while**

---

### 3.2 SEQUENTIAL TARGET RANKING SELECTION METHOD

As mentioned in the introduction, for multi-class and untargeted attacks, the widely used and reliable method is to minimize the loss function $-(z_y - z_t)$ for each target $t \neq y$, and then take the most adversarial one. Since the computational cost of such a solution depends on the number of classes, it is unacceptable for practitioners with limited computational resources. Here, we propose a fast solution which saves a large amount of running time with little-to-no performance lost. We call our solution the *Sequential TArget Ranking Selection* (STARS), which consists of the following strategies.

Given a natural input $x$ and a $K$-class classifier $f$, denoting the predicted probability as $f(x)_i$ for a false label $i$, a natural intuition is that the false target $i$ with a higher value of $f(x)_i$ is more likely to lead to a successful attack. To verify this intuition, in Figure 3, we compare the performance between only selecting the false targets with $K_s$ highest predicted probabilities and selecting all the $K - 1$ false targets. The results show that only selecting $K_s$ (e.g., $K_s = 3$) highest targets achieves comparable performance. Combining this strategy with MM attack, we then denote MM3 (or MM5) as the MM attack with $K_s = 3$ (or $K_s = 5$). For the ranking of predicted probability $f(x)_i$, we also investigate the difference of replacing the natural input $x$ with adversarial examples in Appendix D, which shows that the replacement only has limited improvements. For the sake of efficiency, we recommend to use the ranking of $f(x)_i$ of the natural input $x$, which does not need extra computation.

Figure 3 also motivates us to perform a sequential attack based on the ranking of predicted probability on false targets: first, consider the false target $i$ with the highest predicted probability $f(x)_i$; if the attack succeeds, terminate MM attack on other targets; otherwise, continue to consider the false target with the second highest predicted probability, and so on.

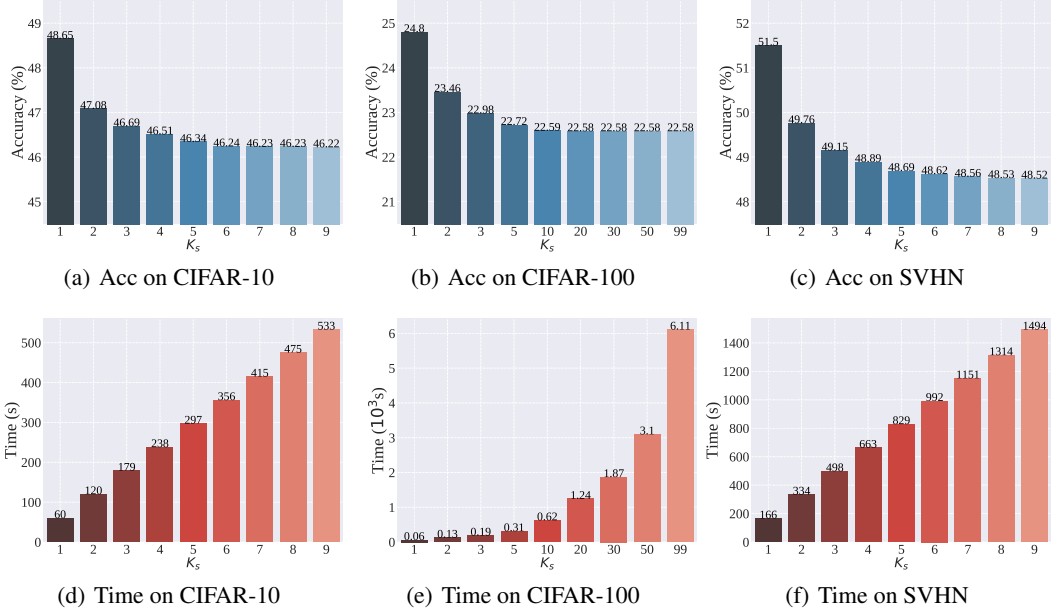

(a) Acc on CIFAR-10     (b) Acc on CIFAR-100     (c) Acc on SVHN

(d) Time on CIFAR-10     (e) Time on CIFAR-100     (f) Time on SVHN

Figure 3: Reliability and computational time of different selection number. Regarding the performance for all targets as benchmarks, as shown in the subfigure (a), (b) and (c), pre-selecting 3 (or 5) targets can achieve 99.13% (or 99.78%) reliability on CIFAR-10, 99.49% (or 99.82%) reliability on CIFAR-100; 98.78% (or 99.67%) reliability on SVHN; as shown in the subfigure (d), (e) and (f), pre-selecting 3 (or 5) targets only costs 33.58% (or 55.72%) computational time on CIFAR-10, 3.11% (or 5.07%) computational time on CIFAR-100; 33.33% (or 55.49%) computational time on SVHN.

Our experiments show that with the help of STARS method, MM3 attack saves 76.36% of the computational time on CIFAR-10, 98.51% on CIFAR-100 and 77.78% on SVHN.

### 3.3 MINIMUM-MARGIN ATTACK

With the above considerations, we summarize our scheme of MM attack in Algorithm 1. We follow the setting of the adaptive step size selection in Croce & Hein (2020) and specify checkpoints $W = \{w_0 = 0, ..., w_n\}$ at which the MM attack decides whether to halve the current step size. The two conditions in Algorithm 1 are:

$$\text{Condition 2.} \quad \sum_{i=w_{j-1}}^{w_j - 1} 1_{f(x'_{i+1}) > f(x'_i)} < \beta \cdot (w_j - w_{j-1}), \tag{9}$$

$$\text{Condition 3.} \quad \alpha^{w_{j-1}} \equiv \alpha^{w_j} \text{ and } f_{max}^{w_{j-1}} \equiv f_{max}^{w_j}. \tag{10}$$

## 4 EXPERIMENTS

In this section, we justify the efficacy of our MM attack. In the experiments, we consider $\ell_\infty$-norm bounded perturbation (i.e., $||x' - x||_\infty \le \epsilon$) in both training and evaluations. The images of all datasets are normalized into [0,1].

### 4.1 BASELINES

We evaluate 3 versions of our MM attack and compare them with 8 baselines. MM3 (or MM5) is the version of MM attack with maximum step number $K = 20$ and targets selection number $K_s = 3$ (or $K_s = 5$); MM+ is the version of MM attack with maximum step number $K = 100$ and targets selection number $K_s = 9$. Baselines consist of PGD (Madry et al., 2018), CW (Carlini & Wagner, 2017b), APGD-CE, APGD-DLR, FAB, Square, AA and T-AA (Croce & Hein, 2020).

Table 3: Test accuracy (%) of adversarial training.

| Methods | PGD | Diff. | CW | Diff. | MM3-F10 | Diff. | MM3-F20 | Diff. | MM3 | Diff. |
|---|---|---|---|---|---|---|---|---|---|---|
| PGD (Test) | 51.14 | -4.10 | 51.47 | -3.77 | 54.96 | -0.28 | **55.24** | 0.00 | 55.04 | -0.20 |
| CW (Test) | 49.95 | -1.89 | **53.26** | 0.00 | 51.18 | -2.08 | 51.16 | -2.10 | 51.84 | -1.42 |
| A-CE (Test) | 48.58 | -3.92 | 48.16 | -4.34 | 51.55 | -0.95 | **52.50** | 0.00 | 52.22 | -0.28 |
| A-DLR (Test) | 48.85 | -1.44 | **52.76** | 0.00 | 49.78 | -2.98 | 49.88 | -2.88 | 50.29 | -2.47 |
| FAB (Test) | 47.28 | -1.22 | 47.13 | -1.37 | 47.83 | -0.67 | 48.28 | -0.22 | **48.50** | -0.00 |
| Square (Test) | 54.46 | -0.66 | **55.32** | 0.00 | 54.80 | -0.52 | 54.83 | -0.49 | 55.12 | -0.20 |
| AA (Test) | 46.43 | -1.85 | 46.36 | -1.92 | 47.62 | -0.66 | 47.84 | -0.44 | **48.28** | -0.00 |
| T-AA (Test) | 46.12 | -0.97 | 45.26 | -1.83 | 46.39 | -0.70 | 46.73 | -0.36 | **47.09** | -0.00 |
| MM3 (Test) | 46.69 | -1.17 | 46.77 | -1.09 | 47.20 | -0.66 | 47.48 | -0.38 | **47.86** | -0.00 |
| MM9 (Test) | 46.21 | -0.95 | 45.36 | -1.80 | 46.49 | -0.67 | 46.82 | -0.34 | **47.16** | -0.00 |
| MM+ (Test) | 46.12 | -0.90 | 45.22 | -1.80 | 46.39 | -0.63 | 46.68 | -0.34 | **47.02** | -0.00 |

## 4.2 EXPERIMENTAL SETUP

We verify our methods on different neural networks including ResNet-18 (He et al., 2016) and Wide-ResNet-34 (WRN-34) (Zagoruyko & Komodakis, 2016) using different datasets: CIFAR-10, CIFAR-100, SVHN. The *adversarial training* (AT) method in this paper follows (Madry et al., 2018). The training setup follows previous works (Madry et al., 2018; Zhang et al., 2019) that all networks are trained for 100 epochs using SGD with 0.9 momentum. The initial learning rate is 0.1 (0.01 for SVHN), and is divided by 10 at epoch 60 and 90, respectively. The weight decay is 0.0002 (0.0035 for SVHN). The previous work (Rice et al., 2020) observed that overfitting in robust adversarial training hurts test set performance. Thus, following Rice et al. (2020), we compare different methods based on the performance of the best checkpoint model (the early stopping results at epoch 60). For generating the adversarial data for updating the network, we set the $\ell_\infty$-norm bounded perturbation $\epsilon_{train} = 8/255$; the maximum number of PGD steps is $K = 10$; step size $\alpha = \epsilon_{train}/10$. In testing, unless otherwise specified, we set $\ell_\infty$-norm bounded perturbation $\epsilon_{test} = 8/255$. For PGD and CW, we follow the setting in (Zhang et al., 2020a): the maximum number of steps $K = 20$, and the step size $\alpha = \epsilon/4$. For APGD-CE, APGD-DLR, FAB and their corresponding targeted version, we follow the setting in (Croce & Hein, 2020): the maximum number of steps $K = 100$. There is a random start in training and testing, i.e., uniformly random perturbations ($[-\epsilon_{train}, +\epsilon_{train}]$ and $[-\epsilon_{test}, +\epsilon_{test}]$) are added to natural instances.

## 4.3 PERFORMANCE EVALUATION

We report the performance of our MM attack and all baselines on CIFAR-10, CIFAR-100, SVHN with the model structure chosen as ResNet-18. As shown in Figure 4, first, our MM attack can perform better than any single attack of PGD, CW, A-DLR, A-CE and FAB; second, compared with the ensemble of diverse attacks AA and T-AA, our MM attack achieves comparable performance but only incurs a very small amount of computational time. Extensive experiments on the large-capacity network WRN-34 are provided in Appendix E.

## 4.4 ADVERSARIAL TRAINING WITH MM ATTACK

By injecting adversarial examples into the training data, *adversarial training* (AT) methods seek to train an adversarial-robust deep neural network whose predictions are locally invariant in a small neighborhood of its inputs. Existing empirical defense methods formulate the adversarial training as a min-max optimization problem (Madry et al., 2018). Since a large number of adversarial examples need to be generated during training, practitioners pay great attention to the time cost of adversarial

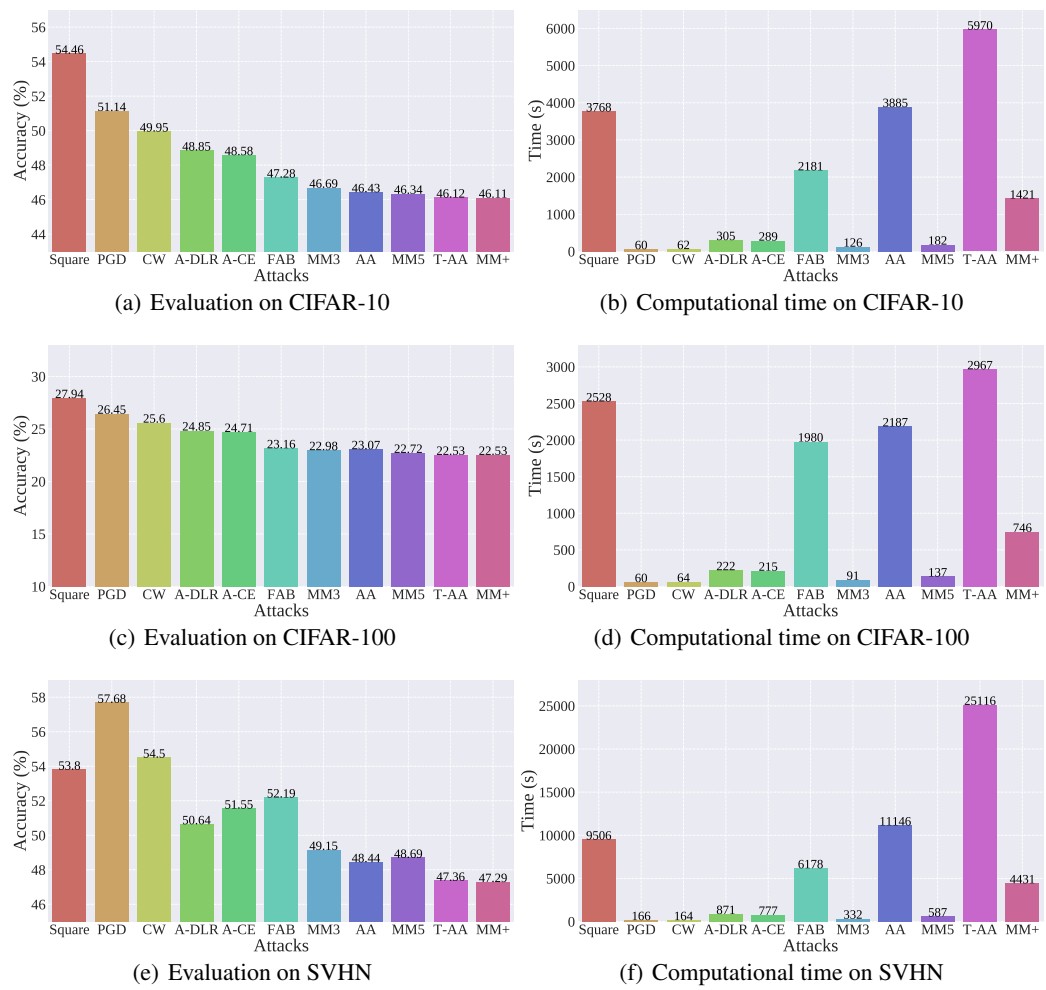

Figure 4: Comparison of reliability and computational cost. We compare three versions of our MM attack (MM3, MM5 and MM+ mentioned in Section 4.1) with 8 baselines. In subfigure (a), (c) and (e), the Y-axis is the accuracy of the attacked model, which means that the lower the accuracy, the stronger the attack (or to say the better evaluation). In subfigure (b), (d) and (f), the Y-axis is computational time, which means the less the time, the higher the computational efficiency.

example generation. Compared with AA, MM attack contributes a feasible and reliable method to generate high-quality adversarial examples in AT. Table 3 reports the performance. MM3-F10 (MM3-F20) denotes AT using adversarial examples generated by MM attack with 10 (20) fixed steps. MM3 denotes AT using adversarial examples generated by MM attack with 20 adaptive steps. We choose 11 testing methods to evaluate the robustness of 5 adversarial trained models. As shown in Table 3, by replacing PGD generated adversarial examples with MM attack generated adversarial examples, the robustness of AT model is significantly improved.

## 5 CONCLUSION

In this work, we proposed MM attack, which can reliably and efficiently evaluate adversarial robustness. For its reliability, we identified minimum-margin (MM) as the key evaluation criterion for the most adversarial example. For its computational efficiency, we proposed an effective STARS method to ensure that its computational time is independent of the number of classes. Our experiments showed that MM attack achieves comparable performance compared with AA, but only costs 3% of the computational time. Its reliability and efficiency further allow us to extend MM attack into AT, which significantly improves the quality of adversarial examples in AT.

ETHICS STATEMENT

This paper does not raise any ethics concerns. This study does not involve any human subjects, practices to data set releases, potentially harmful insights, methodologies and applications, potential conflicts of interest and sponsorship, discrimination/bias/fairness concerns, privacy and security issues, legal compliance, and research integrity issues.

REPRODUCIBILITY STATEMENT

To ensure the reproducibility of experimental results, we will provide a link for an anonymous repository about the source codes of this paper in the discussion phase.

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

## A    THE COMPUTATIONAL TIME IN THE WORST CASE

In this section, we discuss the computational time in the worst case. According to their attack mechanism as an ensemble of diverse attacks, AA and T-AA consider one attack first. If the attack succeeds, stop other attacks on the current example; else, continue to consider the next attack in the ensemble. According to the strategy of our STARS method, MM attack considers the false target with the largest predicted probability first, if the attack succeeds, stop attacks on other false targets; else, continue to consider the next target in the ranking of the predicted probability. The computational time of these methods is influenced by different datasets and models. Hence, in the worst case that all attacks inside fail to succeed, the computational time is the sum of the individual time of each attack. Hence, the computational cost of AA (or T-AA) is 109 times (or 440 times ) more than PGD, and 34 times (or 139 times) more than MM3 in this case.

## B    THE REALIZATION OF ADVERSARIAL TRAINING OF MM ATTACK

We summarize the adversarial training of MM Attack in Algorithm 2. We use MM3 attack to generate adversarial examples, and the computational time is about 2 times as much as PGD (Madry et al., 2018), which can be acceptable for most practitioners.

## C    POTENTIAL BENEFITS OF DIVERSE RESCALINGS

We investigate the difference among different successful sets of seven rescaling methods mentioned above. In Table 2, the setting follows (Madry et al., 2018) (with 20 fixed steps). In Table 4, the setting follows (Croce & Hein, 2020) (with 100 adaptive steps). In Table 2 and Table 4, the non-empty difference sets $A \cup B_i - A$ and $A \cup B_i - B_i$ suggest that diverse rescaling methods can complement each other. Hence, when considerable computational resources are available, we recommend practitioners to consider diverse logits rescaling on a strong attack (e.g., our MM attack) rather than diverse weak attacks. Note that we do not argue that diverse weak attacks is unnecessary but rather that when a reliable enough attack exists, most relatively weak attacks have limited benefits other than increased computational cost.

## D    THE REPLACEMENT OF NATURAL DATA FOR THE RANKING IN STARS

In our STARS method, we also investigate the difference of replacing the natural input $x$ with adversarial examples. Table 5 shows that the replacement has limited improvements.

## E    DETAILED EXPERIMENTAL RESULTS

To verify the rationality of minimum-margin, we conduct experiments on different step size, different step number and different $\mathcal{B}_\epsilon[x]$ in Table 6 and Table 7. We compare the reliability and the computational time between MM attacks and baselines. In Table 8 and Table 9, unless specified, the model structure is ResNet-18. The experiments verify that our MM attack achieves comparable performance but only incurs a very small amount of computational time.

## F    EXPERIMENTAL RESOURCES

We implement all methods on Python 3.7 (Pytorch 1.7.1) with an NVIDIA GeForce RTX 3090 GPU with AMD Ryzen Threadripper 3960X 24 Core Processor. The CIFAR-10 dataset, the SVHN and the CIFAR-100 dataset can be downloaded via Pytorch. Given the $50,000$ images from the CIFAR-10 and CIFAR-100 training set, $73,257$ digits from the SVHN training set, we conduct the adversarial training on ResNet-18 and Wide ResNet-34 for classification.

---

**Algorithm 2** Adversarial Training of MM attack.

---

1: **Input:** network architecture $f$ parametrized by $\theta$, training dataset $S$, loss function $l$, learning rate $\eta$, number of epochs $T$, batch size $n$;
2: **Output:** Adversarial robust network $f_\theta$;
3: **for** epoch = $1, 2, \ldots, T$ **do**
4:     **for** mini-batch = 1,2,...,$N$ **do**
5:         Sample a mini-batch $\{(x_i, y_i)\}_{i=1}^n$ from $S$;
6:         **for** $i = 1, 2, \ldots, n$ **do**
7:             Obtain adversarial data of MM attack $x'_i$ of $x_i$ by Algorithm 1;
8:         **end for**
9:         $\theta \leftarrow \theta - \eta \sum_{i=1}^n \nabla_\theta \ell\left(f_\theta\left(x'_i\right), y_i\right)/n$;
10:     **end for**
11: **end for**

---

Table 4: The successful set of different rescaling methods.

| ID | Rescaling method | Formulation | Successful set | Ranking | diff. | $A \cup B_i - A$ | $A \cup B_i - B_i$ |
|----|------------------|-------------|----------------|---------|-------|------------------|--------------------|
| A | Natural logits | $-(z_y - z_t)$ | 5379 | 1 | 0 | 0 | 0 |
| $B_1$ | Softmax | $-\frac{e^{z_y} - e^{z_t}}{\sum_{i=0}^K e^{z_i}}$ | 5377 | 2 | -2 | 3 | 5 |
| $B_2$ | Max | $-\frac{z_y - z_t}{z_y}$ | 5374 | =4 | -5 | 3 | 8 |
| $B_3$ | Sum | $-\frac{z_y - z_t}{z_y + z_t}$ | 5374 | =4 | -5 | 3 | 8 |
| $B_4$ | Min-Max | $-\frac{z_y - z_t}{z_{\pi_1} - z_{\pi_{10}}}$ | 5376 | 3 | -3 | 4 | 7 |
| $B_5$ | DLR | $-\frac{z_y - z_t}{z_{\pi_1} - \frac{1}{2} \cdot (z_{\pi_3} + z_{\pi_4})}$ | 5372 | 6 | -7 | 2 | 9 |
| $B_6$ | Sigmoid | $-(\frac{e^{z_y}}{1+e^{z_y}} - \frac{e^{z_t}}{1+e^{z_t}})$ | 5311 | 7 | -68 | 2 | 70 |

Table 5: Test accuracy (%): Replacing natural data with adversarial data in STARS method.

| Dataset | Reference attack | Select-$\epsilon$ | MM3 | Diff. | MM9 | Diff. |
|---------|------------------|-------------------|-----|-------|-----|-------|
| CIFAR-10 | None | 8/255 | 48.23 | -0.42 | 47.81 | 0.00 |
| CIFAR-10 | FGSM | 8/255 | 48.05 | -0.24 | 47.81 | 0.00 |
| CIFAR-10 | PGD-20 | 8/255 | **47.92** | -0.11 | 47.81 | 0.00 |
| CIFAR-10 | PGD-20 | 6/255 | 47.98 | -0.17 | 47.81 | 0.00 |
| CIFAR-10 | PGD-20 | 4/255 | 48.04 | -0.23 | 47.81 | 0.00 |
| SVHN | None | 8/255 | 52.45 | -0.61 | 51.84 | 0.00 |
| SVHN | FGSM | 8/255 | 52.07 | -0.23 | 51.84 | 0.00 |
| SVHN | PGD-20 | 8/255 | **51.97** | -0.13 | 51.84 | 0.00 |
| SVHN | PGD-20 | 6/255 | 52.00 | -0.16 | 51.84 | 0.00 |
| SVHN | PGD-20 | 4/255 | 52.07 | -0.23 | 51.84 | 0.00 |
| CIFAR-100 | None | 8/255 | 23.92 | -0.41 | 23.51 | 0.00 |
| CIFAR-100 | FGSM | 8/255 | 23.63 | -0.12 | 23.51 | 0.00 |
| CIFAR-100 | PGD-20 | 8/255 | **23.57** | -0.06 | 23.51 | 0.00 |
| CIFAR-100 | PGD-20 | 6/255 | **23.57** | -0.06 | 23.51 | 0.00 |
| CIFAR-100 | PGD-20 | 4/255 | 23.63 | -0.12 | 23.51 | 0.00 |

Table 6: Test accuracy (%): the rationality of MM under different step sizes and step numbers.

| Step size | Step num | PGD-20 | Diff. | CW | Diff. | MM3-F | Diff. | MM9-F | Diff. |
|---|---|---|---|---|---|---|---|---|---|
| CIFAR-10 | | | | | | | | | |
| 0.003 | 20 | 51.14 | -3.33 | 49.95 | -2.14 | 48.23 | -0.42 | **47.81** | 0.00 |
| 1/255 | 40 | 50.16 | -3.15 | 49.13 | -2.12 | 47.46 | -0.45 | **47.01** | 0.00 |
| 1/255 | 20 | 50.28 | -3.22 | 49.19 | -2.13 | 47.50 | -0.44 | **47.06** | 0.00 |
| 1/255 | 40 | 49.30 | -2.92 | 48.45 | -2.07 | 46.88 | -0.50 | **46.38** | 0.00 |
| 2/255 | 10 | 50.54 | -3.26 | 49.38 | -2.10 | 46.70 | -0.42 | **47.28** | 0.00 |
| 2/255 | 20 | 49.36 | -2.93 | 48.48 | -2.05 | 46.92 | -0.49 | **46.43** | 0.00 |
| 4/255 | 10 | 49.52 | -2.97 | 48.60 | -2.05 | 47.02 | -0.47 | **46.55** | 0.00 |
| SVHN | | | | | | | | | |
| 0.003 | 20 | 57.68 | -5.84 | 54.42 | -2.58 | 52.45 | -0.61 | **51.84** | 0.00 |
| 1/255 | 40 | 56.03 | -5.78 | 52.90 | -2.65 | 50.91 | -0.66 | **50.25** | 0.00 |
| 1/255 | 20 | 56.81 | -5.74 | 53.69 | -2.62 | 51.72 | -0.65 | **51.07** | 0.00 |
| 1/255 | 40 | 55.49 | -5.50 | 52.59 | -2.60 | 50.65 | -0.66 | **49.99** | 0.00 |
| 2/255 | 10 | 57.30 | -5.71 | 54.12 | -2.53 | 52.19 | -0.60 | **51.59** | 0.00 |
| 2/255 | 20 | 55.45 | -5.32 | 52.70 | -2.57 | 50.79 | -0.66 | **50.13** | 0.00 |
| 4/255 | 10 | 56.16 | -5.13 | 53.52 | -2.49 | 51.62 | -0.59 | **51.03** | 0.00 |

Table 7: Test accuracy (%): the rationality of MM under different $\mathcal{B}_\epsilon[x]$.

| $\epsilon$ | PGD-20 | Diff. | CW | Diff. | MM3-F | Diff. | MM9-F | Diff. |
|---|---|---|---|---|---|---|---|---|
| ResNet-18 | | | | | | | | |
| 4 | 67.90 | -0.70 | 68.06 | -0.86 | 67.23 | -0.03 | **67.20** | 0.00 |
| 8 | 51.14 | -3.33 | 49.95 | -2.14 | 48.23 | -0.42 | **47.81** | 0.00 |
| 12 | 45.53 | -4.62 | 43.85 | -2.94 | 41.86 | -0.95 | **40.91** | 0.00 |
| WRN-34 | | | | | | | | |
| 4 | 70.23 | -0.30 | 70.55 | -0.62 | 69.94 | -0.01 | **69.93** | 0.00 |
| 8 | 53.69 | -2.07 | 53.89 | -2.27 | 51.95 | -0.33 | **51.62** | 0.00 |
| 12 | 46.76 | -3.68 | 46.24 | -3.16 | 44.05 | -0.97 | **43.08** | 0.00 |

Table 8: Evaluation: test accuracy (%) on different datasets and model structures.

| Methods | CIFAR-10 | Diff. | CIFAR-100 | Diff. | SVHN | Diff. | [WRN34] CIFAR-10 | Diff. |
|---|---|---|---|---|---|---|---|---|
| PGD | 51.14 | -5.03 | 26.45 | -3.92 | 57.68 | -10.39 | 53.70 | -3.88 |
| CW | 49.95 | -3.84 | 25.60 | -3.07 | 54.50 | -7.21 | 53.90 | -4.08 |
| A-CE | 48.58 | -2.47 | 24.71 | -2.18 | 51.55 | -4.26 | 51.00 | -1.18 |
| A-DLR | 48.85 | -2.74 | 24.85 | -2.32 | 50.64 | -3.35 | 52.24 | -2.42 |
| FAB | 47.28 | -1.17 | 23.16 | -0.63 | 52.19 | -4.90 | 51.04 | -1.22 |
| Square | 54.46 | -8.35 | 27.94 | -5.41 | 53.80 | -6.51 | 58.04 | -8.22 |
| AA | 46.43 | -0.32 | 23.07 | -0.54 | 48.44 | -1.15 | 50.21 | -0.39 |
| T-AA | 46.12 | -0.01 | 22.53 | 0.00 | 47.36 | -0.07 | 49.82 | 0.00 |
| MM3 | 46.69 | -0.58 | 22.98 | -0.45 | 49.15 | -1.86 | 50.26 | -0.44 |
| MM5 | 46.34 | -0.23 | 22.72 | -0.19 | 48.69 | -1.40 | 49.99 | -0.17 |
| MM+ | 46.11 | 0.00 | 22.53 | 0.00 | 47.29 | 0.00 | 49.82 | 0.00 |

Table 9: Evaluation: the computational time (s) on different datasets and model structures.

| Methods | CIFAR-10 | Diff. | CIFAR-100 | Diff. | SVHN | Diff. | [WRN34] CIFAR-10 | Diff. |
|---|---|---|---|---|---|---|---|---|
| PGD | 60 | 0 | 60 | 0 | 166 | -2 | 416 | -10 |
| CW | 62 | -2 | 64 | -4 | 164 | 0 | 406 | 0 |
| A-CE | 289 | -229 | 215 | -155 | 777 | -613 | 1910 | -1504 |
| A-DLR | 305 | -245 | 222 | -162 | 871 | -707 | 1901 | -1495 |
| FAB | 2181 | -2121 | 1980 | -1920 | 6178 | -6014 | 13809 | -13403 |
| Square | 3768 | -3708 | 2528 | -2468 | 9506 | -9342 | 22593 | -22187 |
| AA | 3885 | -3825 | 2187 | -2127 | 11146 | -10982 | 29637 | -29231 |
| T-AA | 5970 | -5910 | 2967 | -2907 | 25116 | -24952 | 40178 | -39772 |
| MM3 | 126 | -66 | 91 | -31 | 332 | -168 | 796 | -390 |
| MM5 | 182 | -122 | 137 | -77 | 587 | -423 | 1342 | -936 |
| MM+ | 1421 | -1361 | 746 | -686 | 4431 | -4267 | 10773 | -10367 |

