# OpenReview forum: "Fast and Reliable Evaluation of Adversarial Robustness with Minimum-Margin Attack"
_ICLR.cc/2022/Conference — ICLR 2022 Submitted_

### Official Review · Reviewer_BW92 · 2021-10-30

**Correctness:** 2
**Technical Novelty And Significance:** 2
**Empirical Novelty And Significance:** 2
**Recommendation:** 3
**Confidence:** 4

**Main Review:**

### Comments:

- The threat model is not stated anywhere. There is no definition of adversarial robustness/robust accuracy. It is not clear then if (1) the attack is a minimum-distance or maximum-confidence attack, (2) it is a targeted or untargeted attack in the common sense used in the field (as opposed to the "targeted" version of APGD, that is instead just using the "targets" for reducing the number of adversarial classes to consider in the optimization), and (3) if the attack is only defined in the $\ell_\infty$ norm.


### Flaws in the experimental evaluation

- the evaluation does not consider a state-of-the-art attack such as [brendel2020]. This is a pity, as [brendel2020] is presented as a "fast and reliable method" for evaluating robustness, and has similar desired characteristics as this attack.
- The parameters of the attacks used seem sub-optimal. There is no choice of the hyperparameters, and using 10 steps for PGD seems to be limiting the capabilities of the attack. The same can be true for the CW attack, especially as there is no mention on how many binary-search steps are being used. It is OK to test the attacks with limited resources, but a more detailed asymptotic analysis (e.g. with 1k steps) would concretely support the claims of the paper that the attack remains comparable to the other attacks while reducing computational time.
- The runtimes are computed in an uneven scenario. The total time per-step, or better, per-query to the model should be used instead of the total cumulative time. This makes no sense. As an alternative, one should compare the capabilities of the PGD attack (depicted here as fast but not reliable) with a fixed computational time, i.e., by increasing the number of steps performed by PGD until it spends the same amount of time as the MM attack.
- the authors did not state if they used some available implementations of the attack, or implemented their own versions. Since the computational time depends on the implementation, this might be a problem when using the runtime as a benchmark.

### Incorrect statements and unsupported claims

**Abstract**

- there is no definition of the "most adversarial example", even though there are several references of this in the paper. This is also used in the abstract. Depending on the objective, a stong adversarial example can be seen in different ways. I suggest to expand this with a definition.
- there is no evidence suggesting that the PGD attack is 100 times slower than AA. The comparison is performed in uneven scenario, where AA uses 100 iterations while PGD uses 10. Moreover, this is stated in the abstract, which makes the statement easy to take and quote, without knowing the context. This statement should be removed.


**Introduction**

- "for practitioners who need real-time evaluation at each epoch of the training process of a robust model". Is there real cases that require this kind of evaluation? This is missing a reference.
- "Unfortunately, PGD fails to reliably evaluate adversarial robustness of a robust DNN". This sentence is over-generalistic and not true for the majority of the cases. PGD was succesfully used against many defenses, just by making it adaptive to the defense [tramer2020].
- "CE loss, which is based on the probability of the true label $p_y$, is not an appropriate measure to the quality of adversarial examples". There is no definition in the paper for "quality of adversarial examples", which makes this statement very confusing.
- "Hence, the reliable method is to minimize $z_y - z_t$ for each $t \neq y$ and take the most adversarial one, which is a widely used solution". This is not widely-used, as for now it seems only used in [croce2020].

**Preliminary**

- "$x^{(0)}$ refers to the starting point which corresponds to the natural example (or the natural example perturbed by a small Gaussian or uniformly random noise)". The statement within parentheses makes the definition of the closed ball in eq. 2 makes the ball centered in $x^{(0)}$. This does not correspond to the adversarial robustness measured in the original clean sample.
- many equations (see Eqs. 3-7) depend on f, x, y, but they often don't appear inside the equations.
- "They showed that using adaptive step size significantly improves the adversarial robustness". Should be "improves the adversarial examples" or "improves the adversarial evaluation". The attacks are not improving robustness.

**Realization**

- Eqs. 8 and 9 use variables ($\alpha$, $\beta$) never introduced in the text.


### Minor issues

- the comparison with targeted-dlr loss in sect. 3 should be clarified. It is very difficult to read, and it does not really capture the advantage of using different methods for rescaling. This might be better supported by some evidence or toy example, and surely by adding some insight on which the hypothesis is based on. Moreover, the authors should then explain what is the difference from the CW loss, as it seems that they are using that one.
- Figures and tables need descriptive captions that clarify what is being depicted. In particular, tables need improvements in the headers and some highlighting of the results. It is also a good practice to mention them in the text (Figure 1b). Figure 2 is difficult to understand, and contains a legend with unclear definitions (see "classified area"). In table 2, it is impossible to understand what are the values presented in the cells. The algorithm needs some hints/comments/description.


### References:

- [tramer2020] Tramer, Florian, et al. "On Adaptive Attacks to Adversarial Example Defenses." Advances in Neural Information Processing Systems 33 (2020).

- [croce2020] Croce, Francesco, and Matthias Hein. "Reliable evaluation of adversarial robustness with an ensemble of diverse parameter-free attacks." International conference on machine learning. PMLR, 2020.

- [brendel2020] Brendel, W., et al. "Accurate, reliable and fast robustness evaluation." Thirty-third Conference on Neural Information Processing Systems (NeurIPS 2019). Curran, 2020.

**Summary Of The Paper:**

The paper proposes an attack for testing adversarial robustness that is reportedly faster than the state-of-the-art attacks but still produces reliable results. The advantage in speed is obtained by using a sequential target ranking selection method, while reliability is achieved by using a minimum-margin loss.


**Summary Of The Review:**

### Strengths:
- Tries to improve efficiency of adversarial attacks

### Weaknesses:
- the paper is missing some definition that should not be taken for granted
- the evaluation is not entirely convincing and might be unfair
- results should be presented better, as they are very difficult to read and understand

---

### Official Review · Reviewer_Zhx4 · 2021-10-31

**Correctness:** 3
**Technical Novelty And Significance:** 2
**Empirical Novelty And Significance:** 2
**Recommendation:** 3
**Confidence:** 4

**Main Review:**

Strengths:
- This paper is well-written, especially with detailed descriptions and empirical results on the effects of different attacking loss functions.
- The improvements shown in Figure 1 seems promising with significant saving on computation.

Weaknesses:
- Although several attacking baselines are considered, they are all only evaluated against PGD-AT (Madry et al. 2018). This could cause a biased evaluation of the attacking performance. Namely, as a potential substitute for AA, the proposed MM should be widely tested against different defenses, just as done in the AA paper (Croce & Hein, 2020). This should not be computationally hard, considering the efficiency of MM and many existing defenses (and their checkpoints) provided in, e.g., RobustBench.
- The multi-target attacking strategy has already been proposed in [a], but I'm surprised that [a] is not even cited in this paper. For me, the proposed STAR strategy is just a top-K variant for the original multi-target attack. Besides, using logits rather than softmax outputs is also not a new discovery since Carlini & Wagner (2017b). Thus, the technical contribution and novelty of MM are quite limited.

Minors:
- What is the definition of $l\_{MM}$ in Algorithm 1?

References:
[a] Gowal et al. An alternative surrogate loss for pgd-based adversarial testing, 2019.

**Summary Of The Paper:**

This paper proposes a minimum-margin (MM) attack to evaluate defenses. The authors report detailed results on the effects of different loss functions. Experiments are done on CIFAR-10/100 and SVHN, against the adversarially trained models.

**Summary Of The Review:**

Limited technical contribution, lack of evaluations against more defenses.

---

### Official Review · Reviewer_pCJy · 2021-11-02

**Correctness:** 3
**Technical Novelty And Significance:** 4
**Empirical Novelty And Significance:** 3
**Recommendation:** 8
**Confidence:** 4

**Main Review:**

Major contributions:

The main idea has been illustrated in Figure 2. Traditional PGD attack minimizes the probability of the true label (by maximizing the loss), and the proposed minimum margin attack minimizes the margin of the probabilities between the true label and the most confusing label. To the best of my knowledge, the idea is novel in adversarial machine learning.

The computational efficiency of the MM attack is amazing. Nowadays, researchers are still using mainly PGD for training but AA for evaluation because AA is more than 100 times slower than PGD. The MM attack is 20 or even 30 times faster than the AA attack, making it possible for training with stronger adversarial examples besides faster evaluation of the adversarial robustness of given deep learning models. In my opinion, the results are significant.

Concerns:

The paper lacks some theoretical analysis, for example, how would the attack converge, how to guarantee the minimum probability margin example is stronger than the minimum probability example (and thus more informative for both evaluating and training), and whether the iterative MM attack algorithm is as stable as the PGD and AA algorithms.

The experiments mainly focused on the evaluation part and then the training part is quite weak. Although researchers believe stronger adversarial examples lead to more robust models, it is not always the case because adversarial examples can be generated by quite different underlying principles as minimum probability vs. minimum probability margin. It is better to concretely show that MM is almost as fast as PGD and almost as strong as AA for training besides for evaluation. This is quite critical for the significance of the paper.

**Summary Of The Paper:**

The paper proposed a strong adversarial attack, i.e., an attack that can generate strong adversarial examples and thus can better evaluate the adversarial robustness of given deep learning models. Compared with the SOTA attack, the proposed attack is much faster and thus easier to be applied in practice. The idea is novel and the results are solid.


**Summary Of The Review:**

This is an overall well-executed paper, with good novelty and solid experiments. Some points should be clarified and stregnthened in the revision.

---

### Official Review · Reviewer_s3fo · 2021-11-09

**Correctness:** 3
**Technical Novelty And Significance:** 2
**Empirical Novelty And Significance:** 2
**Recommendation:** 5
**Confidence:** 4

**Main Review:**

Strengths:

-- The paper is well-written and the preliminaries are described clearly.

-- The proposed method presents significantly low computational complexity.

Weaknesses:

-- The proposed method is only compared against PGD and CW.

-- The authors have mentioned that: "For reliability, we evaluate the quality of adversarial examples using the margin between two targets for precisely identifying the most adversarial example.". Can you please explain about "most adversarial example" ?

-- $\beta$ in Equation 9 is not defined.

**Summary Of The Paper:**

The authors propose minimum-margin (MM) attack to provide comparable performance with AutoAttack while significantly decreasing the computational cost. They propose Sequential TArget Ranking Selection (STARS) to make the computational cost independent of the number of classes.

**Summary Of The Review:**

Although, I believe the proposed method has the potential for a good publication, I do not recommend the acceptance of the paper in the current form.

---

### Decision · Program_Chairs · 2022-01-20

**Decision:**

Reject

**Comment:**

The paper focuses on the strong adversarial attack, i.e., an attack that can generate strong adversarial examples and thus can better evaluate the adversarial robustness of given deep learning models. One review gave a score of 8 while the other 3 reviewers gave negative scores. The main issue lies in the limited experiments, as a potential substitute for AA, the proposed MM should be widely tested against different defenses, just as done in the AA paper. The writing of the paper is somehow is not rigorous including many incorrect statements and unsupported claims which should be well addressed in the revision. Thus, it cannot be accepted to ICLR for its current version.